# Practical Data-Dependent Metric Compression with Provable Guarantees

**Piotr Indyk**[*]
MIT

**Ilya Razenshteyn**[*]
MIT

**Tal Wagner**[*]
MIT

## Abstract

We introduce a new distance-preserving compact representation of multi-dimensional point-sets. Given $n$ points in a $d$-dimensional space where each coordinate is represented using $B$ bits (i.e., $dB$ bits per point), it produces a representation of size $O(d \log(dB/\epsilon) + \log n)$ bits per point from which one can approximate the distances up to a factor of $1 \pm \epsilon$. Our algorithm almost matches the recent bound of [6] while being much simpler. We compare our algorithm to Product Quantization (PQ) [7], a state of the art heuristic metric compression method. We evaluate both algorithms on several data sets: SIFT (used in [7]), MNIST [11], New York City taxi time series [4] and a synthetic one-dimensional data set embedded in a high-dimensional space. With appropriately tuned parameters, our algorithm produces representations that are comparable to or better than those produced by PQ, while having provable guarantees on its performance.

## 1   Introduction

Compact distance-preserving representations of high-dimensional objects are very useful tools in data analysis and machine learning. They compress each data point in a data set using a small number of bits while preserving the distances between the points up to a controllable accuracy. This makes it possible to run data analysis algorithms, such as similarity search, machine learning classifiers, etc, on data sets of reduced size. The benefits of this approach include: (a) reduced running time (b) reduced storage and (c) reduced communication cost (between machines, between CPU and RAM, between CPU and GPU, etc). These three factors make the computation more efficient overall, especially on modern architectures where the communication cost is often the dominant factor in the running time, so fitting the data in a single processing unit is highly beneficial. Because of these benefits, various compact representations have been extensively studied over the last decade, for applications such as: speeding up similarity search [3, 5, 10, 19, 22, 7, 15, 18], scalable learning algorithms [21, 12], streaming algorithms [13] and other tasks. For example, a recent paper [8] describes a similarity search software package based on one such method (Product Quantization (PQ)) that has been used to solve very large similarity search problems over billions of point on GPUs at Facebook.

The methods for designing such representations can be classified into *data-dependent* and *data-oblivious*. The former analyze the whole data set in order to construct the point-set representation, while the latter apply a fixed procedure individually to each data point. A classic example of the data-oblivious approach is based on randomized dimensionality reduction [9], which states that any set of $n$ points in the Euclidean space of arbitrary dimension $D$ can be mapped into a space of dimension $d = O(\epsilon^{-2} \log n)$, such that the distances between all pairs of points are preserved up to a factor of $1 \pm \epsilon$. This allows representing each point using $d(B + \log D)$ bits, where $B$ is the number

---

[*]Authors ordered alphabetically.

of bits of precision in the coordinates of the original pointset. [2] More efficient representations are possible if the goal is to preserve only the distances in a certain range. In particular, $O(\epsilon^{-2} \log n)$ *bits* are sufficient to distinguish between distances smaller than 1 and greater than $1 + \epsilon$, independently of the precision parameter [10] (see also [16] for kernel generalizations). Even more efficient methods are known if the coordinates are binary [3, 12, 18].

Data-dependent methods compute the bit representations of points "holistically", typically by solving a global optimization problem. Examples of this approach include Semantic Hashing [17], Spectral Hashing [22] or Product Quantization [7] (see also the survey [20]). Although successful, most of the results in this line of research are empirical in nature, and we are not aware of any worst-case accuracy vs. compression tradeoff bounds for those methods along the lines of the aforementioned data oblivious approaches.

A recent work [6] shows that it is possible to combine the two approaches and obtain algorithms that adapt to the data while providing worst-case accuracy/compression tradeoffs. In particular, the latter paper shows how to construct representations of $d$-dimensional pointsets that preserve all distances up to a factor of $1 \pm \epsilon$ while using only $O((d + \log n) \log(1/\epsilon) + \log(Bn))$ bits per point. Their algorithm uses hierarchical clustering in order to group close points together, and represents each point by a displacement vector from a near by point that has already been stored. The displacement vector is then appropriately rounded to reduce the representation size. Although theoretically interesting, that algorithm is rather complex and (to the best of our knowledge) has not been implemented.

**Our results.** The main contribution of this paper is QuadSketch (QS), a *simple* data-adaptive algorithm, which is both provable and practical. It represents each point using $O(d \log(dB/\epsilon) + \log n)$ bits, where (as before) we can set $d = O(\epsilon^{-2} \log n)$ using the Johnson-Lindenstrauss lemma. Our bound significantly improves over the "vanilla" $O(dB)$ bound (obtained by storing all $d$ coordinates to full precision), and comes close to bound of [6]. At the same time, the algorithm is quite simple and intuitive: it computes a $d$-dimensional quadtree[3] and appropriately prunes its edges and nodes.[4]

We evaluate QuadSketch experimentally on both real and synthetic data sets: a SIFT feature data set from [7], MNIST [11], time series data reflecting taxi ridership in New York City [4] and a synthetic data set (Diagonal) containing random points from a one-dimensional subspace (i.e., a line) embedded in a high-dimensional space. The data sets are quite diverse: SIFT and MNIST data sets are de-facto "standard" test cases for nearest neighbor search and distance preserving sketches, NYC taxi data was designed to contain anomalies and "irrelevant" dimensions, while Diagonal has extremely low intrinsic dimension. We compare our algorithms to Product Quantization (PQ) [7], a state of the art method for computing distance-preserving sketches, as well as a baseline simple uniform quantization method (Grid). The sketch length/accuracy tradeoffs for QS and PQ are comparable on SIFT and MNIST data, with PQ having higher accuracy for shorter sketches while QS having better accuracy for longer sketches. On NYC taxi data, the accuracy of QS is higher over the whole range of sketch lengths . Finally, Diagonal exemplifies a situation where the low dimensionality of the data set hinders the performance of PQ, while QS naturally adapts to this data set. Overall, QS performs well on "typical" data sets, while its provable guarantees ensure robust performance in a wide range of scenarios. Both algorithms improve over the baseline quantization method.

## 2 Formal Statement of Results

**Preliminaries.** Let $X = \{x_1, \ldots, x_n\} \subset \mathbb{R}^d$ be a pointset in Euclidean space. A compression scheme constructs from $X$ a bit representation referred to as a *sketch*. Given the sketch, and without access to the original pointset, one can *decompress* the sketch into an approximate pointset

$\tilde{X} = \{\tilde{x}_1, \ldots, \tilde{x}_n\} \subset \mathbb{R}^d$. The goal is to minimize the size of the sketch, while approximately preserving the geometric properties of the pointset, in particular the distances and near neighbors.

In the previous section we parameterized the sketch size in terms of the number of points $n$, the dimension $d$, and the bits per coordinate $B$. In fact, our results are more general, and can be stated in terms of the *aspect ratio* of the pointset, denoted by $\Phi$ and defined as the ratio between the largest to smallest distance,

$$\Phi = \frac{\max_{1 \le i < j \le n} \|x_i - x_j\|}{\min_{1 \le i < j \le n} \|x_i - x_j\|}.$$

Note that $\log(\Phi) \le \log d + B$, so our bounds, stated in terms of $\log \Phi$, immediately imply analogous bounds in terms of $B$.

We will use $[n]$ to denote $\{1, \ldots, n\}$, and $\tilde{O}(f)$ to suppress polylogarithmic factors in $f$.

**QuadSketch.**    Our compression algorithm, described in detail in Section 3, is based on a randomized variant of a quadtree followed by a pruning step. In its simplest variant, the trade-off between the sketch size and compression quality is governed by a single parameter $\Lambda$. Specifically, $\Lambda$ controls the pruning step, in which the algorithm identifies "non-important" bits among those stored in the quadtree (i.e. bits whose omission would have little effect on the approximation quality), and removes them from the sketch. Higher values of $\Lambda$ result in sketches that are longer but have better approximation quality.

**Approximate nearest neighbors.**    Our main theorem provides the following guarantees for the basic variant of QuadSketch: for each point, the distances from that point to all other points are preserved up to a factor of $1 \pm \epsilon$ with a constant probability.

**Theorem 1.** *Given $\epsilon, \delta > 0$, let $\Lambda = O(\log(d \log \Phi / \epsilon \delta))$ and $L = \log \Phi + \Lambda$. QuadSketch runs in time $\tilde{O}(ndL)$ and produces a sketch of size $O(nd\Lambda + n \log n)$ bits, with the following guarantee: For every $i \in [n]$,*

$$\Pr\left[\forall_{j \in [n]} \|\tilde{x}_i - \tilde{x}_j\| = (1 \pm \epsilon)\|x_i - x_j\|\right] \ge 1 - \delta.$$

*In particular, with probability $1 - \delta$, if $\tilde{x}_{i^*}$ is the nearest neighbor of $\tilde{x}_i$ in $\tilde{X}$, then $x_{i^*}$ is a $(1 + \epsilon)$-approximate nearest neighbor of $x_i$ in $X$.*

Note that the theorem allows us to compress the input point-set into a sketch and then decompress it back into a point-set which can be fed to a black box similarity search algorithm. Alternatively, one can decompress only specific points and approximate the distance between them.

For example, if $d = O(\epsilon^{-2} \log n)$ and $\Phi$ is polynomially bounded in $n$, then Theorem 1 uses $\Lambda = O(\log \log n + \log(1/\epsilon))$ bits per coordinate to preserve $(1 + \epsilon)$-approximate nearest neighbors.

The full version of QuadSketch, described in Section 3, allows extra fine-tuning by exposing additional parameters of the algorithm. The guarantees for the full version are summarized by Theorem 3 in Section 3.

**Maximum distortion.**    We also show that a recursive application of QuadSketch makes it possible to approximately preserve the distances between *all* pairs of points. This is the setting considered in [6]. (In contrast, Theorem 1 preserves the distances from any single point.)

**Theorem 2.** *Given $\epsilon > 0$, let $\Lambda = O(\log(d \log \Phi / \epsilon))$ and $L = \log \Phi + \Lambda$. There is a randomized algorithm that runs in time $\tilde{O}(ndL)$ and produces a sketch of size $O(nd\Lambda + n \log n)$ bits, such that with high probability, every distance $\|x_i - x_j\|$ can be recovered from the sketch up to distortion $1 \pm \epsilon$.*

Theorem 2 has smaller sketch size than that provided by the "vanilla" bound, and only slightly larger than that in [6]. For example, for $d = O(\epsilon^{-2} \log n)$ and $\Phi = \text{poly}(n)$, it improves over the "vanilla" bound by a factor of $O(\log n / \log \log n)$ and is lossier than the bound of [6] by a factor of $O(\log \log n)$. However, compared to the latter, our construction time is nearly linear in $n$. The comparison is summarized in Table 1.

Table 1: Comparison of Euclidean metric sketches with maximum distortion $1 \pm \epsilon$, for $d = O(\epsilon^{-2} \log n)$ and $\log \Phi = O(\log n)$.

| REFERENCE | BITS PER POINT | CONSTRUCTION TIME |
|---|---|---|
| "Vanilla" bound | $O(\epsilon^{-2} \log^2 n)$ | – |
| Algorithm of [6] | $O(\epsilon^{-2} \log n \ \log(1/\epsilon))$ | $\tilde{O}(n^{1+\alpha} + \epsilon^{-2}n)$ for $\alpha \in (0, 1]$ |
| Theorem 2 | $O(\epsilon^{-2} \log n \ (\log \log n \ + \log(1/\epsilon)))$ | $\tilde{O}(\epsilon^{-2}n)$ |

We remark that Theorem 2 does not let us recover an approximate embedding of the pointset, $\tilde{x}_1, \ldots, \tilde{x}_n$, as Theorem 1 does. Instead, the sketch functions as an oracle that accepts queries of the form $(i, j)$ and return an approximation for the distance $\|x_i - x_j\|$.

## 3 The Compression Scheme

The sketching algorithm takes as input the pointset $X$, and two parameters $L$ and $\Lambda$ that control the amount of compression.

**Step 1: Randomly shifted grid.** The algorithm starts by imposing a randomly shifted axis-parallel grid on the points. We first enclose the whole pointset in an axis-parallel hypercube $H$. Let $\Delta' = \max_{i \in [n]} \|x_1 - x_i\|$, and $\Delta = 2^{\lceil \log \Delta' \rceil}$. Set up $H$ to be centered at $x_1$ with side length $4\Delta$. Now choose $\sigma_1, \ldots, \sigma_d \in [-\Delta, \Delta]$ independently and uniformly at random, and shift $H$ in each coordinate $j$ by $\sigma_j$. By the choice of side length $4\Delta$, one can see that $H$ after the shift still contains the whole pointset. For every integer $\ell$ such that $-\infty < \ell \leq \log(4\Delta)$, let $G_\ell$ denote the axis-parallel grid with cell side $2^\ell$ which is aligned with $H$.

Note that this step can be often eliminated in practice without affecting the empirical performance of the algorithm, but it is necessary in order to achieve guarantees for *arbitrary* pointsets.

**Step 2: Quadtree construction.** The $2^d$-ary quadtree on the nested grids $G_\ell$ is naturally defined by associating every grid cell $c$ in $G_\ell$ with the tree node at level $\ell$, such that its children are the $2^d$ grid cells in $G_{\ell-1}$ which are contained in $c$. The edge connecting a node $v$ to a child $v'$ is labeled with a bitstring of length $d$ defined as follows: the $j^{th}$ bit is 0 if $v'$ coincides with the bottom half of $v$ along coordinate $j$, and 1 if $v'$ coincides with the upper half along that coordinate.

In order to construct the tree, we start with $H$ as the root, and bucket the points contained in it into the $2^d$ children cells. We only add child nodes for cells that contain at least one point of $X$. Then we continue by recursing on the child nodes. The quadtree construction is finished after $L$ levels. We denote the resulting edge-labeled tree by $T^*$. A construction for $L = 2$ is illustrated in Figure 1.

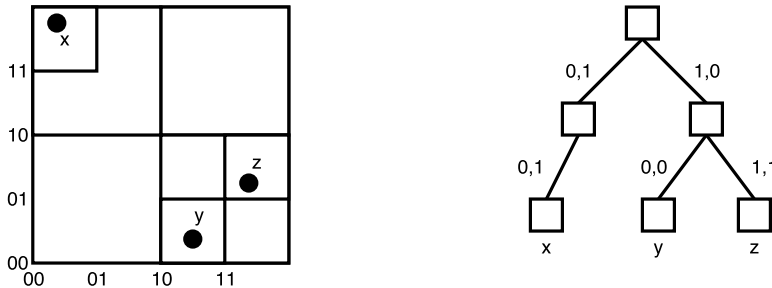

Figure 1: Quadtree construction for points $x, y, z$. The $x$ and $y$ coordinates are written as binary numbers.

We define the *level* of a tree node with side length $2^\ell$ to be $\ell$ (note that $\ell$ can be negative). The *degree* of a node in $T^*$ is its number of children. Since all leaves are located at the bottom level, each point $x_i \in X$ is contained in exactly one leaf, which we henceforth denote by $v_i$.

**Step 3: Pruning.** Consider a downward path $u_0, u_1, \ldots, u_k$ in $T^*$, such that $u_1, \ldots, u_{k-1}$ are nodes with degree 1, and $u_0, u_k$ are nodes with degree other than 1 ($u_k$ may be a leaf). For every such path in $T^*$, if $k > \Lambda + 1$, we remove the nodes $u_{\Lambda+1}, \ldots, u_{k-1}$ from $T^*$ with all their adjacent edges (and edge labels). Instead we connect $u_k$ directly to $u_\Lambda$ as its child. We refer to that edge as the *long edge*, and label it with the length of the path it replaces ($k - \Lambda$). The original edges from $T^*$ are called *short edges*. At the end of the pruning step, we denote the resulting tree by $T$.

**The sketch.** For each point $x_i \in X$ the sketch stores the index of the leaf $v_i$ that contains it. In addition it stores the structure of the tree $T$, encoded using the Eulerian Tour Technique[5]. Specifically, starting at the root, we traverse $T$ in the Depth First Search (DFS) order. In each step, DFS either explores the child of the current node (downward step), or returns to the parent node (upward step). We encode a downward step by 0 and an upward step by 1. With each downward step we also store the label of the traversed edge (a length-$d$ bitstring for a short edge or the edge length for a long edge, and an additional bit marking if the edge is short or long).

**Decompression.** Recovering $\tilde{x}_i$ from the sketch is done simply by following the downward path from the root of $T$ to the associated leaf $v_i$, collecting the edge labels of the short edges, and placing zeros instead of the missing bits of the long edges. The collected bits then correspond to the binary expansion of the coordinates of $\tilde{x}_i$.

More formally, for every node $u$ (not necessarily a leaf) we define $c(u) \in \mathbb{R}^d$ as follows: For $j \in \{1, \ldots, d\}$, concatenate the $j^{th}$ bit of every short edge label traversed along the downward path from the root to $u$. When traversing a long edge labeled with length $k$, concatenate $k$ zeros.[6] Then, place a binary floating point in the resulting bitstring, after the bit corresponding to level 0. (Recall that the levels in $T$ are defined by the grid cell side lengths, and $T$ might not have any nodes in level 0; in this case we need to pad with 0's either on the right or on the left until we have a 0 bit in the location corresponding to level 0.) The resulting binary string is the binary expansion of the $j^{th}$ coordinate of $c(u)$. Now $\tilde{x}_i$ is defined to be $c(v_i)$.

**Block QuadSketch.** We can further modify QuadSketch in a manner similar to Product Quantization [7]. Specifically, we partition the $d$ dimensions into $m$ blocks $B_1 \ldots B_m$ of size $d/m$ each, and apply QuadSketch separately to each block. More formally, for each $B_i$, we apply QuadSketch to the pointset $(x_1)_{B_i} \ldots (x_n)_{B_i}$, where $x_B$ denotes the $m/d$-dimensional vector obtained by projecting $x$ on the dimensions in $B$.

The following statement is an immediate corollary of Theorem 1.

**Theorem 3.** *Given $\epsilon, \delta > 0$, and $m$ dividing $d$, set the pruning parameter $\Lambda$ to $O(\log(d \log \Phi / \epsilon \delta))$ and the number of levels $L$ to $\log \Phi + \Lambda$. The $m$-block variant of QuadSketch runs in time $\tilde{O}(ndL)$ and produces a sketch of size $O(nd\Lambda + nm \log n)$ bits, with the following guarantee: For every $i \in [n]$,*

$$\Pr\left[\forall_{j \in [n]} \|\tilde{x}_i - \tilde{x}_j\| = (1 \pm \epsilon)\|x_i - x_j\|\right] \geq 1 - m\delta.$$

It can be seen that increasing the number of blocks $m$ up to a certain threshold ( $d\Lambda / \log n$ ) does not affect the asymptotic bound on the sketch size. Although we cannot prove that varying $m$ allows to *improve* the accuracy of the sketch, this seems to be the case empirically, as demonstrated in the experimental section.

Table 2: Datasets used in our empirical evaluation. The aspect ratio of SIFT and MNIST is estimated on a random sample.

| Dataset | Points | Dimension | Aspect ratio ($\Phi$) |
|---|---|---|---|
| SIFT | $1,000,000$ | 128 | $\geq 83.2$ |
| MNIST | $60,000$ | 784 | $\geq 9.2$ |
| NYC Taxi | $8,874$ | 48 | 49.5 |
| Diagonal (synthetic) | $10,000$ | 128 | $20,478,740.2$ |

# 4 Experiments

We evaluate QuadSketch experimentally and compare its performance to Product Quantization (PQ) [7], a state-of-the-art compression scheme for approximate nearest neighbors, and to a baseline of uniform scalar quantization, which we refer to as Grid. For each dimension of the dataset, Grid places $k$ equally spaced landmark scalars on the interval between the minimum and the maximum values along that dimension, and rounds each coordinate to the nearest landmark.

All three algorithms work by partitioning the data dimensions into blocks, and performing a quantization step in each block independently of the other ones. QuadSketch and PQ take the number of blocks as a parameter, and Grid uses blocks of size 1. The quantization step is the basic algorithm described in Section 3 for QuadSketch, $k$-means for PQ, and uniform scalar quantization for Grid.

We test the algorithms on four datasets: The SIFT data used in [7], MNIST [11] (with all vectors normalized to 1), NYC Taxi ridership data [4], and a synthetic dataset called Diagonal, consisting of random points on a line embedded in a high-dimensional space. The properties of the datasets are summarized in Table 2. Note that we were not able to compute the exact diameters for MNIST and SIFT, hence we only report estimates for $\Phi$ for these data sets, obtained via random sampling.

The Diagonal dataset consists of $10,000$ points of the form $(x, x, \ldots, x)$, where $x$ is chosen independently and uniformly at random from the interval $[0..40000]$. This yields a dataset with a very large aspect ratio $\Phi$, and on which partitioning into blocks is not expected to be beneficial since all coordinates are maximally correlated.

For SIFT and MNIST we use the standard query set provided with each dataset. For Taxi and Diagonal we use 500 queries chosen at random from each dataset. For the sake of consistency, for all data sets, we apply the same quantization process jointly to both the point set and the query set, for both PQ and QS. We note, however, that both algorithms can be run on "out of sample" queries.

For each dataset, we enumerate the number of blocks over all divisors of the dimension $d$. For QuadSketch, $L$ ranges in $2, \ldots, 20$, and $\Lambda$ ranges in $1, \ldots, L - 1$. For PQ, the number of $k$-means landmarks per block ranges in $2^5, 2^6, \ldots, 2^{12}$. For both algorithms we include the results for all combinations of the parameters, and plot the envelope of the best performing combinations.

We report two measures of performance for each dataset: (a) the *accuracy*, defined as the fraction of queries for which the sketch returns the true nearest neighbor, and (b) the *average distortion*, defined as the ratio between the (true) distances from the query to the reported near neighbor and to the true nearest neighbor. The sketch size is measured in bits per coordinate. The results appear in Figures 2 to 5. Note that the vertical coordinate in the distortion plots corresponds to the value of $\epsilon$, not $1 + \epsilon$.

For SIFT, we also include a comparison with Cartesian k-Means (CKM) [14], in Figure 6.

## 4.1 QuadSketch Parameter Setting

We plot how the different parameters of QuadSketch effect its performance. Recall that $L$ determines the number of levels in the quadtree prior to the pruning step, and $\Lambda$ controls the amount of pruning. By construction, the higher we set these parameters, the larger the sketch will be and with better accuracy. The empirical tradeoff for the SIFT dataset is plotted in Figure 7.

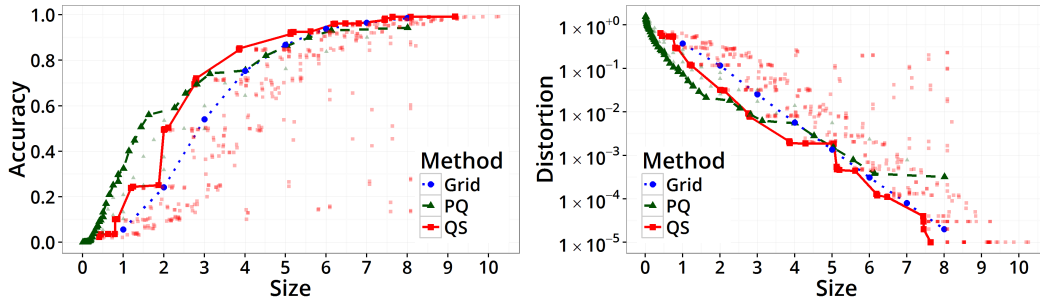

Figure 2: Results for the SIFT dataset.

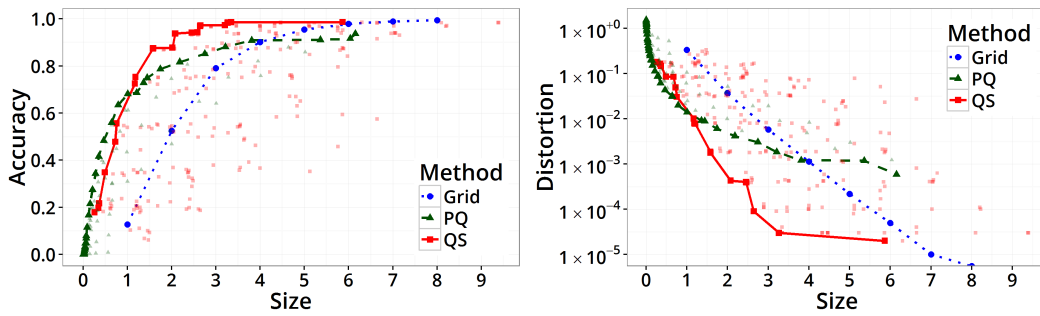

Figure 3: Results for the MNIST dataset.

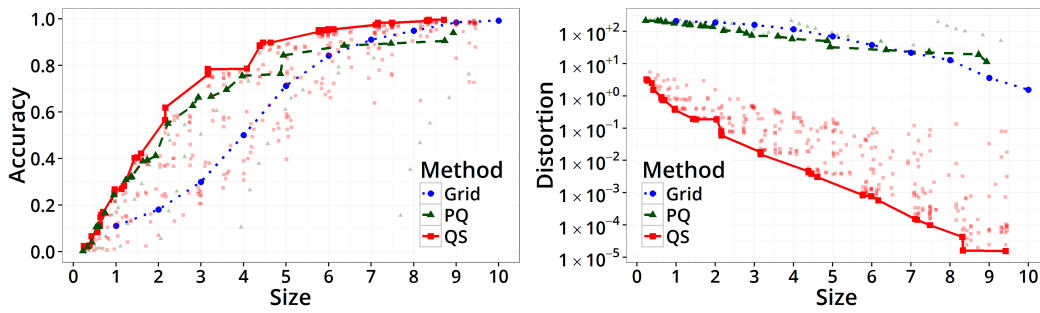

Figure 4: Results for the Taxi dataset.

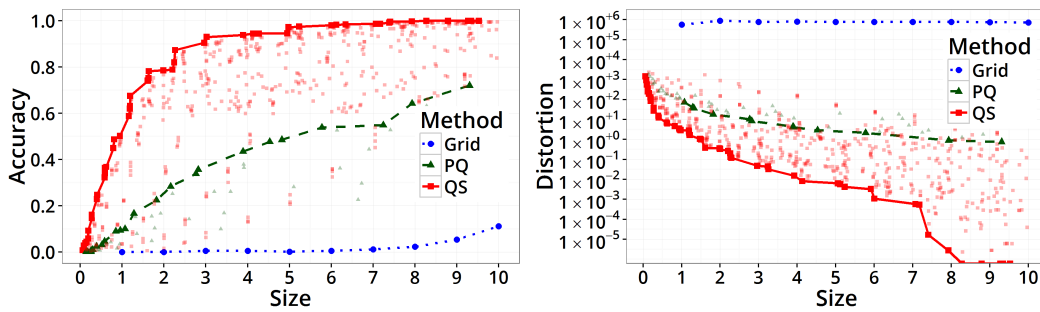

Figure 5: Results for the Diagonal dataset.

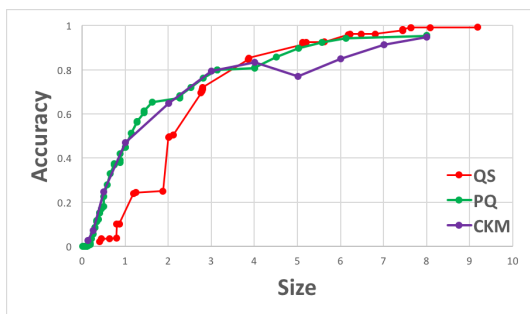

Figure 6: Additional results for the SIFT dataset.

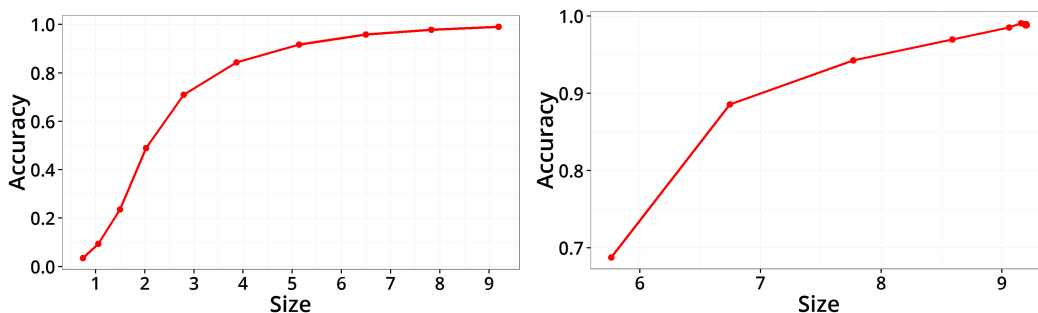

Figure 7: On the left, $L$ varies from 2 to 11 for a fixed setting of 16 blocks and $\Lambda = L - 1$ (no pruning). On the right, $\Lambda$ varies from 1 to 9 for a fixed setting of 16 blocks and $L = 10$. Increasing $\Lambda$ beyond 6 does not have further effect on the resulting sketch.

The optimal setting for the number of blocks is not monotone, and generally depends on the specific dataset. It was noted in [7] that on SIFT data an intermediate number of blocks gives the best results, and this is confirmed by our experiments. Figure 8 lists the performance on the SIFT dataset for a varying number of blocks, for a fixed setting of $L = 6$ and $\Lambda = 5$. It shows that the sketch quality remains essentially the same, while the size varies significantly, with the optimal size attained at 16 blocks.

| # Blocks | Bits per coordinate | Accuracy | Average distortion |
|---|---|---|---|
| 1 | 5.17 | 0.719 | 1.0077 |
| 2 | 4.523 | 0.717 | 1.0076 |
| 4 | 4.02 | 0.722 | 1.0079 |
| 8 | 3.272 | 0.712 | 1.0079 |
| **16** | **2.795** | 0.712 | 1.008 |
| 32 | 3.474 | 0.712 | 1.0082 |
| 64 | 4.032 | 0.713 | 1.0081 |
| 128 | 4.079 | 0.72 | 1.0078 |

Figure 8: QuadSketch accuracy on SIFT data by number of blocks, with $L = 6$ and $\Lambda = 5$.

## Footnotes

[2]The bounds can be stated more generally in terms of the *aspect ratio* $\Phi$ of the point-set. See Section 2 for the discussion.

[3]Traditionally, the term "quadtree" is used for the case of $d = 2$, while its higher-dimensional variants are called " hyperoctrees" [23]. However, for the sake of simplicity, in this paper we use the same term "quadtree" for any value of $d$.

[4]We note that a similar idea (using kd-trees instead of quadtrees) has been earlier proposed in [1]. However, we are not aware of any provable space/distortion tradeoffs for the latter algorithm.

[5]See e.g., https://en.wikipedia.org/wiki/Euler_tour_technique.

[6]This is the "lossy" step in our sketching method: the original bits could be arbitrary, but they are replaced with zeros.

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
