[Supplementary Material]

# 5 Proofs

In this section we prove Theorems 1 and 2. Recall that we have a pointset $x_1, \ldots, x_n \in \mathbb{R}^d$ with aspect ratio $\Phi$, and given error parameters $\epsilon, \delta > 0$. For the remainder of the section we fix the setting

$$\Lambda = \log(16d^{1.5} \log \Phi/(\epsilon\delta)).$$

Recall that in Section 3 we let $G_\ell$ denote the grid with side length $2^\ell$ for every integer $\ell$. Our analysis is based on the observation that randomly shifting the grids, which is used as the first step of our algorithm, induces a *padded decomposition* [2] of the pointset. We now define this formally.

**Definition 1** (padded point). *We say that a point $x_i$ is $(\epsilon, \Lambda, \ell)$-padded, if the grid cell in $G_\ell$ that contains $x_i$ also contains the ball of radius $\rho(\ell)$ centered at $x_i$, where*

$$\rho(\ell) = 8\epsilon^{-1}2^{\ell-\Lambda}\sqrt{d}.$$

*We say that $x_i$ is $(\epsilon, \Lambda)$-padded in the quadtree $T$, if it is $(\epsilon, \Lambda, \ell)$-padded for every level $\ell$ of $T$.*

Note that $d$ and $\Lambda$ are fixed parameters for a given input. We omit their dependence from the notation $\rho(\ell)$ for simplicity.

We now prove Theorem 1. It follows directly from combining the following two lemmas.

**Lemma 1.** *If the grids are randomly shifted, as in Section 3, then every point $x_i$ is $(\epsilon, \Lambda)$-padded in $T$ with probability $1 - \delta$.*

*Proof.* Fix a point $x_i$, a coordinate $k \in \{1, \ldots, d\}$ and a level $\ell$. Let $x_i(k)$ denote the value of $x_i$ in coordinate $k$. Along this coordinate, we are randomly shifting a 1-dimensional grid partitioned into intervals of length $2^\ell$. Since the shift is uniformly random, the probability for $x_i(k)$ to be at distance at most $\rho(\ell)$ from an endpoint of the interval that contains it equals $2\rho(\ell)/2^\ell$. By plugging our setting of $\rho(\ell)$ and $\Lambda$, this probability equals $\delta/(d \log \Phi)$. Taking a union bound over the $d$ coordinates, we have probability at most $\delta/\log \Phi$ for $x_i$ to be at distance at most $\rho(\ell)$ from the boundary of the cell of $G_\ell$ that contains it. In the complement event $x_i$ is $(\epsilon, \Lambda, \ell)$-padded in $G_\ell$. Taking another union bound over the $\log \Phi$ levels in the quadtree, $x_i$ is $(\epsilon, \Lambda)$-padded with probability at least $1 - \delta$. $\square$

**Lemma 2.** *If a point $x_i$ is $(\epsilon, \Lambda)$-padded in $T$, then for every $j \in [n]$,*
$$(1 - \epsilon)\|\tilde{x}_i - \tilde{x}_j\| \leq \|x_i - x_j\| \leq (1 + \epsilon)\|\tilde{x}_i - \tilde{x}_j\|,$$
*where $\{\tilde{x}_i\}$ are as defined in Section 3.*

*Proof.* We recall that $T$ is a pruned quadtree in which every node $v$ is associated with a grid cell of an axis-parallel grid $G_\ell$ with side length $2^\ell$, which is aligned with and contained in $H$. We call $\ell$ the *level* of $v$, and denote it henceforth by $\ell(v)$. We will use the term "bottom-left corner" of a grid cell for the corner that minimizes all coordinate values (i.e., the high-dimensional analog of a bottom-left corner in the plane).

Let $r$ be the root of $T$. We may assume w.l.o.g. that the bottom-left corner of $H$ is the origin in $\mathbb{R}^d$, since translating $H$ together with the entire pointset does not change pairwise distances. Under this assumption, we make the following observation, illustrated in Figure 9.

**Observation 1.** *Let $v$ be a node in $T$. If the path from $r$ to $v$ contains only short edges, then $c(v)$ (defined by the decompression algorithm in Section 3) is the bottom-left corner of the grid cell associated with $v$.*

Let $x_i$ be a padded point, and $x_j$ be any point. Recall that we denote by $v_i$ and $v_j$ the leaves corresponding to $x_i$ and $x_j$ respectively (see Section 3). Let $w$ be the lowest common ancestor of $v_i$ and $v_j$ in $T$. Since $x_i$ and $x_j$ are in separate grid cells of $G_{\ell(w)-1}$, and the cell containing $x_i$ also contains the ball of radius $\rho(\ell(w) - 1)$ around $x_i$, we have

$$\|x_i - x_j\| \geq \rho(\ell(w) - 1) = 8\epsilon^{-1}2^{\ell(w)-1-\Lambda}\sqrt{d}. \tag{1}$$

Let $u_i$ be the lowest node on the downward path from $w$ to $v_i$, that can be reached without traversing a long edge. Similarly define $u_j$ for $v_j$. See Figure 10 for illustration.

Note that $u_i$ must be either the leaf $v_i$, or an internal node whose only outgoing edge is a long edge. In both cases, $u_i$ is the bottom of a path of degree-1 nodes of length $\Lambda$:

Figure 9: By collecting the edge label bits along every dimension from the root to a node, and padding with zeros as necessary, we obtain the binary expansion of the bottom-left corner of the associated grid cell.

- If $u_i$ is a leaf: Since the pointset has aspect ratio $\Phi$, then after $\log \Phi$ levels the grid becomes sufficiently fine such that each grid cell contains at most one point $x_i$. Since we generate the quadtree with $L = \log \Phi + \Lambda$ levels, then each point $x_i$ is in its own grid cell for at least the bottom $\Lambda$ levels of the quadtree.

- If $u_i$ is an internal node which is the head of a long edge: Since the pruning step only places long edges at the bottom of degree-1 paths of length $\Lambda$, then $u_i$ must be the bottom node of such path.

On the other hand $w$ is an ancestor of $u_i$, and it has degree at least 2, since it is also an ancestor of $u_j$. Hence $w$ is at least $\Lambda$ levels above $u_i$, implying $\ell(v_i) \leq \ell(w) - \Lambda$. Applying the same arguments to $u_j$ we get also $\ell(v_j) \leq \ell(w) - \Lambda$.

Figure 10: In the proof of Lemma 2, $w$ is the lowest common ancestor of $v_i, v_j$, the leaves corresponding to $x_i, x_j$. $u_i$ is the lowest node on the downward path from $w$ to $v_i$ which is achievable without traversing any long edges (marked in red). $u_j$ is defined similarly for $v_j$.

Let $c^*(u_i), c^*(u_j) \in \mathbb{R}^d$ be the bottom-left corners of the grid cells associated with $u_i$ and $u_j$. If all edges on the downward paths from the root of $T$ to $u_i$ and $u_j$ were short, then Observation 1 would yield that $c^*(u_i) = c(u_i)$ and $c^*(u_j) = c(u_j)$. In general, there might be some long edges on those paths, but they all must lie on the subpath from the root of $T$ down to $w$, which is the same for

both paths. This is because by the choice of $u_i$ and $u_j$, all downward edges from $w$ to either of them are short. Therefore $c(u_i)$ and $c(u_j)$ are shifted from the true bottom-left corners by the same shift, which we denote by

$$\eta = c^*(u_i) - c(u_i) = c^*(u_j) - c(u_j).$$

Next, observe that the grid cell associated with $u_i$ has side $2^{\ell(u_i)}$ and it contains both $c^*(u_i)$ and $x_j$. Therefore $\|x_i - c^*(u_i)\| \leq 2^{\ell(u_i)}\sqrt{d}$.

Furthermore, since $u_i$ is an ancestor of $v_i$, then by the definition of $c(u_i)$ and $c(v_i)$, in each coordinate, the binary expansions of these two vertices are equal from the location $\ell(u_i)$ and up. In the less significant locations, $c(u_i)$ is zeroed while $c(v_i)$ may have arbitrary bits. This means that the difference between $c(u_i)$ and $c(v_i)$ in each coordinate can be at most $2^{\ell(u_i)}$ in the absolute value, and consequently $\|c(v_i) - c(u_i)\| \leq 2^{\ell(u_i)}\sqrt{d}$. Recalling that the decompression algorithm defines $\tilde{x}_i = c(v_i)$, we get $\|\tilde{x}_i - c(u_i)\| \leq 2^{\ell(u_i)}\sqrt{d}$.

Collecting the above inequalities, we have

$$\begin{aligned}
\|x_i - \eta - \tilde{x}_i\| &= \|x_i - c(u_i) - \eta + c(u_i) - \tilde{x}_i\| \\
&= \|x_i - c^*(u_i) + c(u_i) - \tilde{x}_i\| \\
&\leq \|x_i - c^*(u_i)\| + \|c(u_i) - \tilde{x}_i\| \\
&\leq 2 \cdot 2^{\ell(u_i)}\sqrt{d} \\
&\leq 2 \cdot 2^{\ell(w)-\Lambda}\sqrt{d}.
\end{aligned}$$

Similarly for $j$ we have $\|x_i - \eta - \tilde{x}_i\| \leq 2 \cdot 2^{\ell(w)-\Lambda}\sqrt{d}$. Together, by the triangle inequality,

$$\begin{aligned}
\|\tilde{x}_i - \tilde{x}_j\| &= \|\tilde{x}_i + \eta - x_i + x_i - x_j + x_j - \eta - \tilde{x}_j\| \\
&= \|x_i - x_j\| \pm (\|x_i - \eta - \tilde{x}_i\| + \|x_i - \eta - \tilde{x}_i\|) \\
&= \|x_i - x_j\| \pm 4 \cdot 2^{\ell(w)-\Lambda}\sqrt{d}.
\end{aligned}$$

To complete the proof of Lemma 2 it remains to show $4 \cdot 2^{\ell(w)-\Lambda}\sqrt{d} \leq \epsilon \cdot \|x_i - x_j\|$, which follows from Equation (1). $\qquad\square$

## 5.1 Sketch Size and Running Time

**Lemma 3.** *QuadSketch produces a sketch of size $O(nd\Lambda + n\log n)$ bits.*

*Proof.* The tree $T$ has $n$ leaves, and we have pruned each non-branching path in it to length $\Lambda$. Hence its total size is $O(n\Lambda)$, and its structure can be stored with this many bits using (for example) the DFS scan described in Section 3. Each short edge label is $d$ bits long, so together they consume $O(nd\Lambda)$ bits. As for the long edges, there can be at most $O(n)$ of them, since the bottom of each long edge is either a branching node or a leaf. The long edge labels are lengths of downward paths in the non-pruned tree $T^*$, whose height bounded by is $O(\log \Phi + \Lambda)$. Together the long edge labels consume $O(n\log(\log \Phi + \Lambda))$ bits, which is dominated by $O(n\Lambda)$. Finally for each point $x_i$ we store the index of its corresponding leaf $v_i$, and since there are $n$ leaves, this requires $O(n\log n)$ additional bits to store. $\qquad\square$

**Lemma 4.** *The QuadSketch construction algorithm runs in time $O(ndL)$.*

*Proof.* Given a quadtree cell and a point contained in it, in order to bucket the point into a cell in the next level, we need to check for each coordinate whether the point falls in the upper or lower half of the cell. This takes time $O(d)$. Since each point is bucketed once in every level, and we generate $T^*$ for $L$ levels, the quadtree construction time is $O(ndL)$. The pruning step requires just a linear scan of $T^*$, in time $O(nL)$. $\qquad\square$

## 5.2 Maximum Distortion

We now prove Theorem 2.

**Sketching algorithm**   Given a pointset $X$, apply QuadSketch to $X$ and let $T_1$ be the resulting tree. Let $Q \subset X$ be the padded points in $T_1$ (meaning those for which the condition of Lemma 1 is satisfied for $T_1$). Continue by recursion on $X \setminus Q$, until all points in $X$ are padded in some tree. The returned sketch contains all trees $T_1, \ldots, T_k$ constructed during the recursion, and in addition, for every point $x_i$ we store the index $\gamma_i$ of the tree in which it is padded.

**Query algorithm**   Given two point indices $i, j$, assume w.l.o.g. $\gamma(i) \leq \gamma(j)$, then the tree $T_{\gamma(i)}$ has corresponding leaves for both $x_i$ and $x_j$. We decompress $\tilde{x}_i$ and $\tilde{x}_j$ from $T_{\gamma(i)}$ and return $\|\tilde{x}_i - \tilde{x}_j\|$.

**Analysis**   The correctness of the estimate up to distortion $1 \pm \epsilon$ follows from Lemma 2. We now bound the sketch size and the running time. Lemma 1 with $\delta = 0.25$ implies that in each of the trees $T_1, \ldots, T_k$, the expected fraction of padded points is 0.75. Hence by Markov's inequality, with probability 0.5 at least half the points are padded. Since the calls to QuadSketch are independent (and its success probability depends only on its internal randomness and not on the input points), with probability $0.5^{\lceil \log_2 n \rceil} \sim 1/n$ this happens in each of the first $k = \lceil \log_2 n \rceil$ iterations. This probability can be amplified to constant by $O(\log n)$ independent repetitions. If this event has happened then the sketching algorithm terminates since $Q$ becomes empty. Therefore the total running time of a successful execution is $O(\log^2 n)$ calls to QuadSketch, which by Lemma 4 is $\tilde{O}(ndL)$.

Furthermore, since the number of padded points decreases by at least half in every iteration, the total size of the sketches $T_1, \ldots, T_k$ is

$$O\left( \sum_{k'=0}^{k-1} \frac{n}{2^{k'}} (d\Lambda + \log \frac{n}{2^{k'}}) \right) = O(n(d\Lambda + \log n)),$$

the same as in Theorem 1 up to a constant factor. Finally, since each $\gamma(i)$ is index in $\{1, \ldots, \lceil \log_2 n \rceil\}$, the $\gamma(i)$'s only take additional $O(n \log \log n)$ bits to store. □