[Reviews · NeurIPS 2017]

Reviewer 1



The paper consider the problem of compressing the representation of a set of points in the Euclidean space so that the pairwise distances are preserved and one needs to store as few bits as possible. The problem has been studied in theory but it would be helpful if the authors give more explanation of how this problem is useful in practice. The space needed to store the points is not a barrier in most cases. Additionally, having a data structure that can only give distances between points without anything else is not helpful for tasks that require any kind of indexing e.g. searching for nearest neighbor. The paper gives an algorithm that run faster than previous work but also produce a slightly suboptimal compressed size. The algorithm is basically a version of a compressed quadtree with a small modification: when compressing a long path in the compressed quadtree into a single edge, one does not store the entire description of the path but only the most significant bits (roughly log(1/epsilon) bits for approximation 1+epsilon). Overall, the technical contribution seems limited. However, if the authors can establish this problem as a useful solution for important tasks then the paper would be a nice contribution.

Reviewer 2



This paper presents an impressive algorithm that embeds a set of data points into a small dimensional space, represented by a near-optimal number of bits. The most appealing aspect of this paper is to provide a simple but provable dimensionality reduction algorithm, and supported by empirical experiments. === Strengths === This paper is supported by strong theoretical analysis. And, it is much simpler algorithm than [5] with a slightly worse bit complexity. === Weaknesses === 1. This is an example of data-dependent compression algorithm. Though this paper presents a strong theoretical guarantee, it needs to be compared to state-of-the art data-dependent algorithms. For example, - Optimized Product Quantization for Approximated Nearest Neighbor Search, T. Ge et al, CVPR’13 - Composite Product Quantization for Approximated Nearest Neighbor Search, T. Zhang et al, ICML’14 - And, other recent quantization algorithms. 2. The proposed compression performs worse than PQ when a small code length is allowed, which is the main weakness of this method, in view of a practical side. 3. When the proposed compression is allowed to have the same bit budget to represent each coordinate, the pairwise Euclidean distance is not distorted? Most of compression algorithms (not PCA-type algorithms) suffer from the same problem. === Updates after the feedback === I've read other reviewers' comments and authors' feedback. I think that my original concerns are resolved by the feedback. Moreover, they suggest a possible direction to combine QuadSketch with OPQ, which is appealing.

Reviewer 3



This paper presents a new, practical approach to a field with a huge number of previous papers. They show the potential benefits of their data-adaptive compressive encoding by Accuracy and Distortion experiments on several datasets. When parameters are set correctly for each dataset, the method *can* clearly outperform good methods such as PQ. (So the last sentence of the abstract might be a bit of a strong claim, since sometimes the representations can be worse). The method description is compact, yet readable. It is precise enough that I feel I could sit down, code their algorithm and reproduce their results. It seems, however, that further work is still needed to develop heuristics for adapting the L and Gamma to values appropriate to the dataset. Incorrect settings *might* perform worse than PQ, which at least has decent (smooth) Accuracy/Distortion curves as its parameter (# of landmarks) is varied. I would have liked to see runtime comparison with PQ. It would be nice to have a public "reference implementation" in the future, to spur further development in the community. - line 217: "Section 4.1" should be "Figure 7", which in turn should probably be "Table 3".